# Evaluation of Microaeration and Sound to Increase Biogas Production from Poultry Litter

**John Loughrin *** , **Stacy Antle, Michael Bryant, Zachary Berry and Nanh Lovanh**

United States Department of Agriculture, Agricultural Research Service, Food Animal Environmental Systems Research Unit, 2413 Nashville Road, Suite B5, Bowling Green, KY 42101, USA; stacy.antle@usda.gov (S.A.); mike.bryant@usda.gov (M.B.); zachary.berry2@usda.gov (Z.B.); nanh.lovanh@usda.gov (N.L.)

**\*** Correspondence: john.loughrin@usda.gov; Tel.: +1-270-781-2260

**Abstract:** Microaeration, wherein small amounts of air are introduced into otherwise anaerobic digesters, has been shown to enhance biogas production. This occurs by fostering the growth of facultatively aerobic bacteria and production of enzymes that enhance the degradation of complex polymers such as cellulose. The treatment of anaerobic digestate with sound at sonic frequencies (<20 kHz) has also been shown to improve biogas production. Microaeration at a rate of 800 mL day$^{-1}$, treatment with a 1000-Hz sine wave, and combined microaeration/sound were compared to a control digester for the production of biogas and their effect on wastewater quality. Poultry litter from a facility using wood chips as bedding was used as feed. The initial feeding rate was 400 g week$^{-1}$, and this was slowly increased to a final rate of 2400 g week$^{-1}$. Compared to the control, sound treatment, aeration, and combined sound/aeration produced 17%, 32%, and 28% more biogas. The aeration alone treatment may have been more effective than combined aeration/sound due to the sound interfering with retention of aeration or the formation of free radicals during cavitation. Digesters treated with sound had the highest concentrations of suspended solids, likely due to cavitation occurring within the sludge and the resulting suspension of fine particles by bubbles.

**Keywords:** aeration; anaerobic digestion; biogas; carbon dioxide; cavitation; methane; microaeration

---

## 1. Introduction

Since the widespread adoption of concentrated animal feeding operations (CAFO) in the latter half of the twentieth century, tremendous amounts of animal manures are produced upon limited acreage so that it cannot be disposed of in an environmentally sound and efficient manner. Instead, these wastes are usually treated in facultative anaerobic lagoons, the effectiveness of which varies seasonally, and applied to fields at rates that foster the over accumulation of phosphates and other plant nutrients in soils as well as the runoff of oxygen-consuming organic matter, pathogens, and endocrine-disrupting chemical compounds to waterways [1]. For these reasons, alternative means of handling agricultural wastes are needed to reduce the environmental impacts of CAFO.

The main impediment to implementation of environmentally sound management of agricultural wastes is the economics of food production. Profit margins for producers are often low which limits capital that might be dedicated to infrastructure for waste treatment and nutrient recovery. Conventional wastewater treatment systems such as the activated sludge process are expensive and most often associated with municipal wastewater treatment where loading rates are comparatively low and influent solids are minimal.

Due to these constraints placed on animal waste management, a number of alternate approaches have been studied to reduce the environmental impacts of manure [2]. Among these are waste

compaction, after which waste may be pelletized or bailed. This helps to reduce particulate emissions and facilitates storage and transport prior to land application.

Composting is another process that has received considerable attention as a means of manure handling. In composting, waste is broken down aerobically to reduce and stabilize organic matter as well as reduce pathogens. Considerable losses of ammonia-N occur during composting [3], however, reducing its use as fertilizer although it still has value as a soil conditioner and for its phosphorus content. Due to the high phosphorus concentrations, the composted solids may still need to be transported off site to reduce contamination of ground and surface waters [4].

Anaerobic digestion is another means of potentially reducing the environmental effects of high-strength animal waste. Anaerobic digestion has the potential to reduce sludge volumes, convert organically bound phosphate to ortho-phosphate, making its recovery practical [5], and decrease malodors [6]. The biogas produced by anaerobic digestion can offset the costs associated with waste management or even be a source of income for producers. Biogas production is inherently slow, however, subject to upsets due to overfeeding, and requires considerable investment. Heating and mixing of wastewater are the principal means of dealing with some of these limitations [7,8].

A number of other experimental approaches to increasing the efficiency of anaerobic digestion have been published. Among these are microaeration [9], where small amounts of air are added to otherwise anaerobic digesters. If added in optimal amounts, air can facilitate biogas production by increasing the hydrolysis of sludge and enhancing the growth of facultative acetogens without harming the growth of obligate anaerobes such as methanogenic Archaea and Clostridia [9,10].

Another promising approach is pretreatment of sludge with high-intensity ultrasonic sound to reduce particle size by a combination of vibrational energy and acoustic cavitation [11,12]. By achieving partial disintegration of sludge, as much as a 50% increase in biogas production has been achieved by solubilizing chemical oxygen demand (COD) and facilitating microbial colonization.

As an alternative to ultrasonic pretreatment, we investigated whether continuous in situ treatment of digestate with sound at sonic frequencies (<20 kHz) could also enhance waste breakdown and biogas production. In a first experiment treating waste with single or multiple frequency sine waves, biogas production was increased by approximately 12%, and sludge carbon and nitrogen concentrations were reduced by 19% and 18%, respectively, compared to treatment without sound [13]. In a subsequent pilot-scale experiment employing sine waves, broadband noise, and music to treat anaerobic digestate, biogas production was increased by over 100% compared to the control [14]. It was speculated that the greater enhancement of biogas production in the latter experiment was at least partially due to the larger scale of the experiment eliminating standing acoustical waves of single-frequency sine waves. In addition, the more complex frequencies of broadband noise and music may have also eliminated static pressure maxima and minima within the digester, i.e., a more thorough excitation of the digestate and sludge may have been accomplished by using complex sounds rather than simple sine waves.

We have also investigated whether microaeration would enhance biogas production of poultry litter, a waste product that is normally considered to be a poor candidate for anaerobic digestion due to its high N and lignin content [15]. Microaeration was supplied through a manifold located at the bottom of the digester at 0, 200, 800 or 2000 mL day$^{-1}$ in equally spaced 200-mL increments. Aeration at 200 and 800 mL day$^{-1}$ increased biogas production by 14% and 73%, respectively, while aeration at 2000 mL day$^{-1}$ decreased biogas production by 19%.

In that both microaeration and acoustic treatment of waste were found to enhance biogas production, it seemed natural to investigate whether the two treatments could be combined to realize further benefits for waste reduction and the enhancement of biogas production. Four treatments were compared: a control receiving no sound treatment or microaeration, sound treatment alone, microaeration alone, and microaeration combined with sound treatment. Microaeration at 800 mL day$^{-1}$, supplied in 200-mL increments, was used as this was found to be optimal in previous research [15].

## 2. Materials and Methods

Digesters were constructed from 208-L polyethylene tanks (US Plastic Inc., Lima, OH, USA). The side of each tank had a hole drilled for accepting a 5.08-cm diameter polyvinyl chloride (PVC) pipe fitted with a ball valve that served as a feeding port. The pipe extended into the tank and led to below the surface of the digestate. Float level switches (Omega Engineering Inc., Norwalk, CT, USA) were installed to maintain a digestate volume of 133 L. The float level switch activated an electrical relay (American Zettler, Inc., Aliso Viejo, CA, USA) providing power to a 1.27-cm solenoid-actuated 120 VAC PVC ball valve (Valworx, Inc., Cornelius, NC, USA) installed on a 1.27-cm diameter PVC pipe that served as the waste outlet.

The waste outlet pipe had a hole that housed a 0.3175-cm-diameter Teflon line that led into the inside of the tank and provided aeration to a 2.54-cm-diameter PVC pipe manifold installed in the bottom of the digester. The manifold had an "H" configuration with a volume of approximately 380 mL. The manifold had caps installed on the end of its long arms and a series of 0.3175-cm-diameter holes, allowing communication to the sludge. Aeration was supplied to the subsurface manifold in 200-mL amounts in 15-min intervals four times daily giving an aeration rate of 6.0 mL air per liter of digestate per day total. The air was provided through a 15-cm-tall rotameter at 13.3 mL min$^{-1}$ as supplied by a diaphragm air pump and a 12-volt DC solenoid-actuated gas valve (Spartan Scientific, Boardman, OH, USA) controlled by a rotary timer.

The aeration periods were spaced at equal intervals throughout the day. The cap of each tank was adapted to accommodate a 3-way luer valve and 0.635-cm tubing that served as an outlet and gas sampling port. The tubing was connected to a flowmeter (wettipgasmeter.com) by one arm of the luer valve fitting. The other arm of the fitting was used for taking gas samples by a syringe. The side of the tank had a 0.635-cm diameter port installed for liquid analysis. All pipe connections to the tanks were made with Uniseal® pipe to tank fittings (US Plastic, Inc., Lima, OH, USA).

The speaker in each tank was waterproofed by spray-painting with Rustoleum® spray enamel (Rust-Oleum Corp., Vernon Hills, IL, USA) followed by application of GE Silicone I caulk (General Electric Co., Boston, MA, USA) diluted with petroleum-based charcoal lighter fluid. This increased fluidity of the caulk and ensured more complete coverage of the speakers. Amplification was supplied by a Pyle PTAU45 amplifier rated at 20 W RMS (root-mean-square) power at 1.0 kHz (Pyle Audio, Inc., Brooklyn, NY, USA) with USB memory drive input. During playback of sound to the digester, the amplifier was set to three-quarters volume.

The digesters were set up as follows: one digester was treated as a control, with no supplemental aeration and no sound, one was treated with sound using a 1000 kHz sine wave, one was treated with supplemental aeration and sound, and one was treated with supplemental aeration only.

The poultry litter (PL) had a moisture content of 24.6%. In the remaining dry solids, the volatile solids (VS) content was 77.2% with an ash content of 22.8%. It was obtained from a producer located in South-Central Kentucky. The base material of the litter consisted of wood chips from unspecified *Pinus* species or *Liriodendron tulipfera* L. The digesters were initially 'seeded' with 20 L of liquid obtained from an anaerobic digester on the producer's property. As an initial startup phase, the digesters were fed 400 g PL in 4 L water for seven weeks, giving an initial VS loading rate of 1.75 g VS L$^{-1}$. During this startup phase, biogas concentrations of $CO_2$ and $CH_4$ increased from an average of 12,500 and 3080 μg L$^{-1}$ to 642,000 and 320,000 μg L$^{-1}$, respectively. At this point, the digesters were deemed to have developed a sufficiently mature microbial population to commence treatment of the digesters.

After the startup period, the digesters were successively fed 400 g PL (6 weeks), 500 g PL (6 weeks), 600 g (6 weeks), 700 g PL (6 weeks), 800 g PL (8 weeks), 1 kg (10 weeks), 1.2 kg (7 weeks), 1.6 kg (6 weeks) and 2.4 kg (5 weeks), at which point the amount of accumulated solids in the control treatment made additional feeding difficult and the experiment was ended. PL was suspended in 4 L water and was fed once per week. Feed amounts of 800 g and above were fed twice per week in two equal portions while 2400 g week$^{-1}$ was divided into three equal portions suspended in 4 L of water and fed three times per week.

Gas quality and dissolved gas content were determined by gas chromatography as previously described [13] while gas production was determined with wet tip flowmeters (wettipgasmeter.com). Dissolved carbon and nitrogen were determined on samples taken from digesters from feed loading rates of 800 g PL week$^{-1}$ and above with a Shimadzu Shimadzu TOC-L instrument (Shimadzu Scientific Instruments, Columbia, MD, USA) using APHA standard methods [16] as were other wastewater quality measurements. Statistical analyses of the weekly determinations were performed using SAS version 9.3 (SAS Institute, Cary, NC, USA). Data were analyzed using analysis of variance (ANOVA) of the repeated measurements over time and means compared using Duncan's multiple range test.

## 3. Results and Discussion

### 3.1. Digester Startup

The digesters were seeded with 20 L of digestate from a commercial anaerobic digester until the biogas quality was considered sufficiently good to commence aeration and sound treatments. During the six-week startup period, the pH of the digesters decreased from an average of 6.81 ± 0.015 to 6.49 ± 0.019. However, bicarbonate buffering had increased from an average of 2.64 ± 0.08 mM to 16.9 ± 1.0 mM for all four digesters. As bicarbonate buffering might be considered the single most important water quality component associated with digester stability [17], it was decided to begin air and sound treatments.

Bicarbonate buffering continued to increase after beginning the sound and aeration treatments. At the beginning of the 400 g week$^{-1}$ experimental feeding period, $HCO_3^-$ averaged 17.3 ± 1.27 mM for all treatments and at the end, 67.9 ± 2.0 mM for all four treatments. All four treatments were quite similar in regard to pH and $HCO_3^-$ buffering.

### 3.2. Biogas Production

Likewise, when fed 400 g week$^{-1}$, biogas yields in all four treatments were quite similar (Figure 1). It was only when PL loading rates were increased to 800 g PL week$^{-1}$ (approximately 466 g VS) that the treatments seemed to have a marked effect on biogas production. When fed 800 g week$^{-1}$, air and air plus sound increased biogas production by 46% relative to the control whereas sound treatment alone increased biogas production by only about 3%. As PL feeding rates increased, the sound treatment had more of an effect relative to the control, and the sound only treatment produced 15% more biogas than the control when fed 1000 g week$^{-1}$, and 37% more when fed 1200 g week$^{-1}$. Further information on gas production and wastewater quality in relation to feeding rate can be found as supplementary material (Solids loading rate with averaged data for gas and wastewater quality.xlsx).

The air treatment alone produced more biogas than other treatments for all feeding rates except 400, 800, and 1200 g PL week$^{-1}$, when the combined air and sound treatment produced the most biogas. Thus, the use of microaeration showed the most capacity for increasing biogas production. Table 1 shows the average weekly biogas production for the entire experiment. Compared to the control, sound treatment alone produced 17% more biogas, aeration alone produced 38% more biogas, and combined sound and aeration produced 32% more biogas. The effect was more pronounced at higher feeding rates and when fed 2400 g PL week$^{-1}$, sound treatment produced 26% more biogas than did the control, while aeration and combined sound/aeration produced 65% and 40% more biogas, respectively (Figure 1).

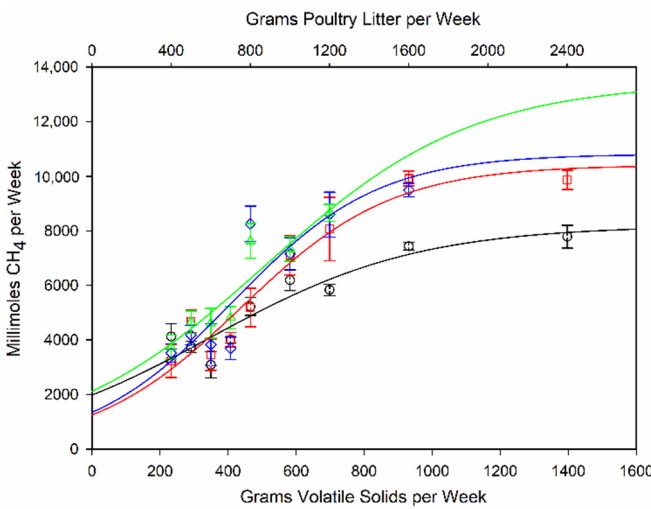

**Figure 1.** Millimoles of methane produced per week. Control: black circles, $r^2 = 0.8808$; digesters treated with sound: red squares, $r^2 = 0.9422$; digesters treated with sound and microaeration: blue diamonds, $r^2 = 0.8702$; digesters treated with microaeration: green triangles, $r^2 = 0.9631$. Data represent the mean ± standard error of the mean.

**Table 1.** Biogas and digestate characteristics of poultry litter digestates.

| Parameter | Treatment | | | |
|---|---|---|---|---|
| | Control | Aeration | Sound | Aeration/Sound |
| | Biogas Characteristics | | | |
| Weekly Biogas (L) [a] | 172 ± 8.79 c | 238 ± 14.8 a | 202 ± 13.0 b | 228 ± 14.2 a |
| Carbon dioxide ($\mu$mole L$^{-1}$) [b] | 14,200 ± 784 a | 13,700 ± 727 a,b | 13,700 ± 2000 b | 13,400 ± 713 b |
| Methane ($\mu$mole L$^{-1}$) [b] | 25,400 ± 1930 a | 25,200 ± 1740 a,b | 24,400 ± 2040 b | 24,400 ± 1730 b |
| | Digestate Characteristics [b] | | | |
| pH | 7.23 ± 0.08 a | 7.22 ± 0.05 b | 7.23 ± 0.06 a | 7.23 ± 0.05 a,b |
| Bicarbonate | 92.9 ± 5.42 a | 91.4 ± 5.41 b | 91.5 ± 5.97 a,b | 89.5 ± 5.31 a,b |
| Solvated carbon dioxide | 10.1 ± 0.39 b | 10.7 ± 0.44 a | 10.1 ± 0.44 b | 10.1 ± 0.41 b |
| Solvated methane | 24.0 ± 1.29 b | 27.7 ± 0.88 a | 24.9 ± 1.16 b | 25.5 ± 0.92 b |
| Chemical oxygen demand | 5070 ± 353 b | 4440 ± 289 d | 5390 ± 350 a | 4580 ± 319 c |
| Total suspended solids | 410 ± 63.0 b | 591 ± 80.4 b | 1360 ± 198 a | 650 ± 120 b |
| Dissolved carbon [c] | 5210 ± 157 a | 4130 ± 158 c | 5290 ± 161 a | 4460 ± 174 b |
| Dissolved nitrogen [c] | 1480 ± 62.6 a | 1050 ± 56.0 d | 1400 ± 56.3 b | 1180 ±58.6 c |

[a] Data represent the mean ± standard error of the mean of 61 weekly gas productions summed from daily determinations. Within a row, means labelled by the same letter are not significantly different by a Duncan's multiple range test at $p = 0.05$. [b] Data represent the mean ± standard error of 61 weekly determinations. Within a row, means labelled by the same letter are not significantly different by a Duncan's multiple range test at $p = 0.05$. [c] Data represent the mean ± standard error of 35 weekly determinations. Within a row, means labelled by the same letter are not significantly different by a Duncan's multiple range test at $p = 0.05$.

Biochemical methane potential (BMP) is a standard means of testing the yield and degradability of feedstocks for anaerobic digestion [18]. Theoretically, BMP reports a maximum potential yield of 0.35 L of CH$_4$ per g of VS [19]. However, BMP tests are best performed in batch measurements [17] as the accumulation of undigested feedstock can affect results. In addition to the complication of the present experiments being performed in semi-batch mode, the digesters were kept in a greenhouse maintained at 26.7 °C, near the midpoint for mesophilic digestion. The relatively low temperature of the experiment helped ensure accumulation of poultry litter.

The control treatment had its greatest apparent yield of CH$_4$ at 0.24 L g$^{-1}$ VS early in the experiment when fed 400 g PL week$^{-1}$. The sound treatment had its greatest yield at 0.37 L CH$_4$ g$^{-1}$ VS when being fed 500 g PL week$^{-1}$. Both the sound-treated and sound- and air-treated digesters yielded

0.42 L $CH_4$ $g^{-1}$ VS when fed 800 g $week^{-1}$. Gas yields had a negative trend relative to feeding rate and when fed 2400 g $week^{-1}$, $CH_4$ yields for the control, sound-treated, sound- and air-treated, and air-treated digesters were 0.13, 0.25, 0.18, and 0.21 L $CH_4$ $g^{-1}$ VS, respectively. At the conclusion of the experiment, the relative yield of $CH_4$ for all feeding rates averaged 0.24 ± 0.27, 0.28 ± 0.016, 0.30 ± 0.025, and 0.32 ± 0.021 mL $CH_4$ $g^{-1}$ VS for the control, sound-treated, sound and air-treated, and air-treated digesters, respectively. From these results, it can be seen that aeration, sound, and sound and aeration combined all made the digesters more capable of handling higher loading rates and that little benefit from these treatments was realized at lower feeding rates. This is clearly illustrated in Figure 1.

### 3.3. Wastewater Quality

None of the treatments had a pronounced effect on chemical oxygen demand (Figure 2). Aeration alone reduced it the most, but COD increased rapidly with increased waste loading rates. COD tests typically overestimate the concentration of soluble organic substances available for anaerobic digestion. A large portion of the soluble matter remaining after anaerobic digestion will consists of slowly biodegradable substances that may be resistant even to aerobic degradation [19]. In that wood chips were used as bedding material for the poultry litter, perhaps much of the soluble COD in the digestate consisted of lignans, condensed and hydrolysable tannins, and other low molecular weight phenolics resistant to anaerobic digestion. It is not unexpected, therefore, that dissolved carbon concentrations were lower in the two aeration treatments (Table 1).

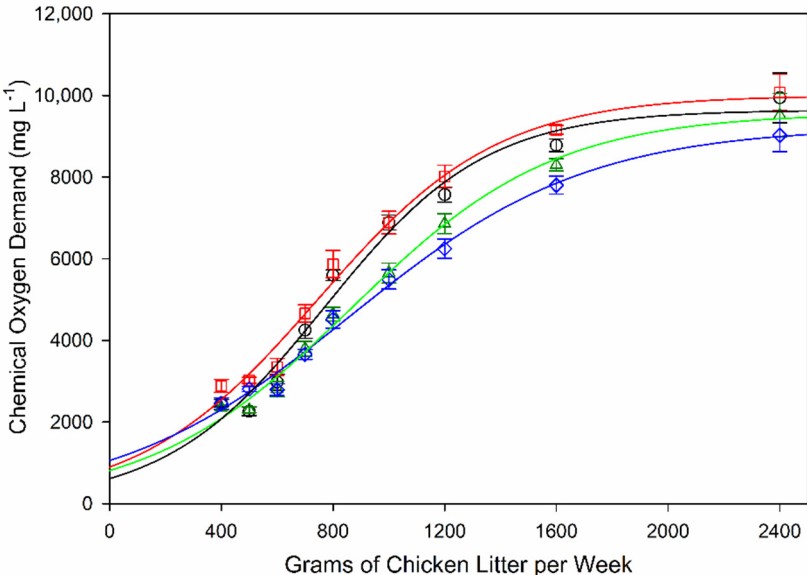

**Figure 2.** Chemical oxygen demand in digestate. Control: black circles, $r^2$ = 0.9426; digesters treated with sound: red squares, $r^2$ = 0.9384; digesters treated with microaeration, $r^2$ = 0.9475: blue diamonds; digesters treated with sound and microaeration: green triangles, $r^2$ = 0.9530.

Thus, given the nature of the bedding material used for PL, it is unsurprising that aeration was more effective than sound in enhancing biogas production. Many aromatic and phenolic compounds are degraded poorly or not at all under anaerobic conditions [20]. While the sound treatment can accelerate waste breakdown by a number of physical mechanisms including vibrational energy imparted to sludge and sludge disruption due cavitation inception and collapse [11,12,14], there is no a priori reason to assume a biological mechanism for accelerated degradation of these refractory molecules except for perhaps stimulation of microbial metabolism [21].

Most other parameters were quite similar in all treatments. Bicarbonate buffering, pH and dissolved $CO_2$ and $CH_4$ were affected very little if at all by the treatments. Still, the concentration

of $CO_2$ and $CH_4$ in the biogas were highest in the control digester, and significantly lower in the two sound treatments. This is in contrast to earlier studies where sound was seen to increase the concentrations of these two compounds [13,14].

Interestingly, the sound treatment had a pronounced effect on suspended solids (Figure 3). The sound only treatment had the highest concentrations of suspended solids at waste loading rates of 500 g PL week$^{-1}$ and above. Fine particles have been found to attach to bubbles formed by cavitation and assist in their flotation [22]. Similarly, biogas recirculation through digesters has been used in a similar manner to increase mixing [23]. While bubbles may be formed by hydrodynamic cavitation in which bubbles form in regions of reduced pressure, as occur in the outlets of restricted flows or in the turbulent wake of propellers, bubbles may also form due to acoustic cavitation in where bubbles grow and oscillate, and eventually collapse, in the presence of an acoustic field [24].

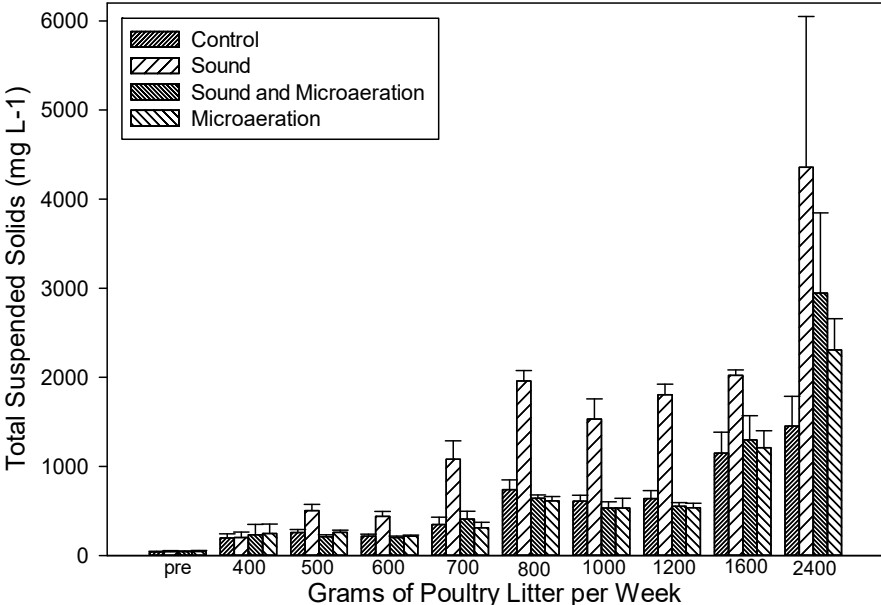

**Figure 3.** Total suspended solids in digestate. Data represent the mean and standard error of the mean.

When sound was combined with aeration, there was no pronounced increase in suspended solids except at a loading rate of 2400 g PL week$^{-1}$ where aeration itself may have also caused an increase in suspended solids. As discussed, there was no benefit to treating the digestate with sound in addition to aeration for biogas production. In previous work, aeration at 800 mL day$^{-1}$ through the subsurface manifold was found to increase biogas production by 73% although with lower PL loading rates [15]. The rationale for introducing aeration to the digestate through a subsurface manifold was that the volume of the manifold (380 mL) would act to retain the aeration within the sludge rather than allowing it to escape, thereby increasing the efficiency of the process and not allowing air to dilute the biogas. The sonic treatment may have interfered with this treatment in that vibrational energy may have caused loss of air from the manifold and may have also fostered the collapse of air bubbles within the manifold. The cavitationally induced collapse of air bubbles has been known to generate free radicals which could be harmful to anaerobic organisms [25].

### 3.4. Ion Analyses

Neither sound or aeration nor sound and aeration combined had a pronounced effect on the concentrations of ions in the digestate (Table 2). Concentration of the monovalent ions potassium, sodium and chloride were lower in digesters receiving microaeration. We know of no reason for this, but it is possible that microaeration allowed for more growth of facultative bacteria and semi-oxygen-tolerant Archaea, as has been previously demonstrated [26]. If microbial growth was

stimulated enough, it is possible that the salinity was lowered somewhat due to the increased microbial biomass and resulting uptake of ions.

**Table 2.** Ion chromatographic analysis of poultry litter digestates.

| Ion | Treatment | | | |
|---|---|---|---|---|
| | **Control** | **Aeration** | **Sound** | **Aeration/Sound** |
| | **Concentration (mg L$^{-1}$) [a]** | | | |
| Phosphate | 13.9 ± 2.2 a | 13.6 ± 2.2 a | 13.7 ± 2.3 a | 14.2 ± 2.4 a |
| Ammonium | 89.0 ± 10.8 a | 90.3 ± 10.1 a | 91.3 ± 10.45 a | 89.1 ± 9.9 a |
| Nitrate | nd b | 0.7 ± 0.1 a | nd b | 0.9 ± 0.2 a |
| Sulfate | 4.3 ± 0.5 a | 4.2 ± 0.6 a | 4.3 ± 0.6 a | 4.2 ± 0.6 a |
| Magnesium | 18.1 ± 1.6 b | 16.7 ± 1.5 c | 19.5 ± 1.7 a | 17.9 ± 1.5 b |
| Calcium | 24.0 ± 3.3 a | 22.8 ± 2.8 a,b | 22.1 ± 3.0 b | 24.3 ± 2.9 a |
| Sodium | 299 ± 31.0 a | 264 ± 25.8 b | 289 ± 28.6 a | 266 ± 26.5 b |
| Potassium | 809 ± 95.9 a | 684 ± 79.2 c | 759 ± 87.0 b | 693 ± 79.4 b |
| Chloride | 667 ± 66.4 a | 580 ± 53.4 c | 630 ± 52.1 b | 577 ± 53.2 c |

[a] Data represent the mean ± standard error of the mean of 61 once-weekly determinations. Means followed by the same letter are not significantly different by a Duncan's multiple range test at $p = 0.05$.

Low concentrations of nitrate were detected in the digesters receiving aeration although the concentrations of ammonium were virtually identical in all four digesters. Still, it seemed some low level of nitrification occurred in the digesters receiving aeration although it could not be expected for nitrate to accumulate in the digestate due to utilization of nitrate by bacteria able to use it as an alternative electron acceptor for respiration [27]. Low activity of nitrification coupled with denitrification in the digesters receiving aeration may help explain the lower concentrations of dissolved nitrogen in these tanks (Table 1) even though there was no significant difference in ammonium levels among the digesters (Table 2).

## 4. Conclusions

Aeration or sound treatment alone significantly enhanced biogas production from PL digestate. Combining the treatments was more effective than sound treatment alone, but not as effective as aeration by itself. Waste treatment is expensive and, in many situations and places, inadequate. Simple technologies like microaeration or sound treatment have the potential to improve wastewater treatment and enhance biogas production. These treatments may be implemented in new or construction or existing structures with little modification and at little cost. This is especially important in agriculture and developing nations where funds for wastewater infrastructure may be wanting.

**Supplementary Materials:** The following are available online at http://www.mdpi.com/2076-3298/7/8/62/s1, Table S1: Solids loading rate with averaged data for gas and wastewater quality.xlsx.

**Author Contributions:** Conceptualization, J.L.; methodology, J.L., S.A., M.B., N.L. and Z.B.; formal analysis, J.L., S.A., M.B. and Z.B.; investigation, S.A.; writing-original draft preparation, J.L.; writing—review and editing, J.L. and S.A.; visualization, J.L., S.A. and M.B.; supervision, J.L. and N.L.; project administration, J.L. All authors have read and agreed to the published version of the manuscript.

**Funding:** This research received no external funding and was conducted as part of USDA-ARS National Program 212: Soil and Water, Developing Safe, Efficient and Environmentally Sound Management Practices for the Use of Animal Manure.

**Acknowledgments:** We thank John McLean of MAC Farms for chicken litter and digestate used to seed the experiment.

**Conflicts of Interest:** The authors declare no conflict of interest.

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
