# Peer review of "Evaluation of Microaeration and Sound to Increase Biogas Production from Poultry Litter"

_environments, doi:10.3390/environments7080062_

Round 1

Reviewer 1 Report

Overall: The paper compared the effect of microaeration, sound, and combination of both on the biogas production of poultry litter. The experimental design was pretty straightforward. There are some grammar issues of the writing that need to be improved.   

Some minor comments:

  1. Why 1000 Hz was chosen for this study?
  2. P112, it was not clear to me how the aeration was controlled. What is the aeration rate per min? for how long?
    “Aeration was supplied to the subsurface manifold in 200 mL increments over 15 min intervals four times daily. “

Does that mean pump 15 min each time, 4 times throughout the day? And the total air volume was 200 mL for each 15 min? Then the aeration rate was 200 mL/15 min =13.3 mL/min over each 15 min? If so, it is confusing to use “increment” and “interval”. It should be rewritten.

  1. P134, Poultry litter (PL) with a moisture content of 24.6% and volatile solids (VS) content of 77.2%. So the TS is 76.4%? how VS is even higher than TS?
  2. P139, it is better to add percentage of CO2 and CH4.
  3. P142, with the increase of organics loading rate, the changes of data along with time, such as CH4, COD, TSS, pH, etc., can be provided as supplementary information.

Author Response

Reviewer 1.

Overall: The paper compared the effect of microaeration, sound, and combination of both on the biogas production of poultry litter. The experimental design was pretty straightforward. There are some grammar issues of the writing that need to be improved.   

Some minor comments:

Overall: The paper compared the effect of microaeration, sound, and combination of both on the biogas production of poultry litter. The experimental design was pretty straightforward. There are some grammar issues of the writing that need to be improved.   

Some minor comments:

  1. Why 1000 Hz was chosen for this study?

1000 Hertz was chosen because this was the frequency used in the previous paper, “Sound enhances wastewater degradation and improves anaerobic digester performance” in which we investigated the used of sound to enhance biogas production. We had found a positive effect of this frequency on biogas production.

  1. P112, it was not clear to me how the aeration was controlled. What is the aeration rate per min? for how long? 
    “Aeration was supplied to the subsurface manifold in 200 mL increments over 15 min intervals four times daily. “

Does that mean pump 15 min each time, 4 times throughout the day? And the total air volume was 200 mL for each 15 min? Then the aeration rate was 200 mL/15 min =13.3 mL/min over each 15 min? If so, it is confusing to use “increment” and “interval”. It should be rewritten.

We did rewrite this to make it clearer.

  1. P134, Poultry litter (PL) with a moisture content of 24.6% and volatile solids (VS) content of 77.2%. So the TS is 76.4%? how VS is even higher than TS? \

We rewrote this to make it clearer. The PL litter had a moisture content of 24.6%. Of the dry matter remaining the VS content was 77.2%, ash 22.8%.

  1. P139, it is better to add percentage of CO2 and CH4.

In Table 1 and Figure 1, we changed reporting CH4 and CO2 concentrations from micrograms to millimolar concentrations. At this point reporting CH4 and CO2 as micrograms per liter gives the reader an appreciation of how much the gas quality improved and we would prefer to leave the numbers as they are. We did list the percentage of CH4 in the supplementary information (see response to next question).

  1. P142, with the increase of organics loading rate, the changes of data along with time, such as CH4, COD, TSS, pH, etc., can be provided as supplementary information.

We have now included an excel file containing the requested supplementary information. Please note that we had to modify the information on the number of observations on dissolved carbon and nitrogen since we neglected to include this previously (last paragraph of Material and Methods, table 1, supplementary information). We apologize for the oversight and have corrected this. The reason for this was that the instrument was purchased and became available at this point during the experiment. We also made an error on the footnote of table 2, where we listed the number of observations as being 39. We corrected this to 61.  

Reviewer 2 Report

Summary: The authors carried out a study in which they investigated the impact of sound waves and microaeration (and their interaction) on anaerobic digestion of poultry litter slurry. They found that sound and microaeration had benefits to biogas production (similar to their past findings) but that there was minimal benefit of interaction. I found the study interesting but a little simplistic. Most of my comments are aimed at improving the clarity of the presentation.

Major comments:

  1. Any time an aeration rate or solids loading rate are reported, it needs to be reported as a specific (normalized by reactor volume) rate to have meaning beyond this study. This occurs throughout the manuscript e.g. L14-15, L17.
  2. No description of analytical methods is provided; only a citation on L150. This forces the reader to go find the other paper to figure out what the authors did. Instead, provide a brief description of the methods and then cite if a reader wants to get more details.
  3. More specifics are needed on statistical analysis – what tests were used? It should also be noted that averages were calculated across repeat measures from the same reactor – i.e. no biological replication. Thus measurements are not truly independent which is an assumption of most statistical tests. Researchers do this all the time due to practical limitations but probably need to point it out.
  4. The results and discussion jumps around quite a bit. Headers within this section would help indicate transitions to new topics.
  5. In Figure 1, I would like to see a second panel graph showing specific biogas production (or even better, CH4) per g VS as a function of loading rate. I suspect that microaeration increases overall biogas but at the expense of specific methane production. This is somewhat confirmed by the values on L197.
  6. Table 1, the presentation of results is a little confusing. In the footer, it says values are the “mean of 61 summed weekly gas production.” I initially took this to mean that total cumulative biogas was summed over the entire period but realized this value is really just the average weekly biogas production across all different solids loading rates. This decision unfortunately hinders the statistical power of the analysis because different loading rates are expected to lead to different total production yet this variation is now part of the error term. Why not instead normalize all production (biogas, CO2, CH4) by the VS loading rate before taking the average? This would correct for the expected variation in biogas production in response to changing VS loading and reduce variability. I understand this rate will still vary with loading rate but I think this is more appropriate than the current presentation of data. In addition, doing so makes the result comparable to the literature since reporting of L CH4/g VS is a standard metric for AD. I also would expect that COD, TSS, soluble C, and soluble N are a function of loading rate and may benefit from normalization before taking an average across loading rates.
  7. Reporting methane and CO2 as ug/L makes it hard to compare to the literature. Normally a molar or volumetric (at STP) ratio is used.

Minor comments:

L142: 400 g PL/week for 6 weeks?

L187: 0.35 ml/g VS does not look right. Should be 0.35 L/g VS. Also, this value will vary widely depending on substrate and digester conditions – the citation references general AD. I would encourage the authors to instead find other studies of poultry litter AD and report that number instead. It would be a much more fair comparison to the author’s work.

Table 2: I’m surprised these ammonium levels are so low given the fairly high PL loadings. TN in Table 1 seems reasonable and is >10x higher than ammonium which makes me think this digester is not very effective at nutrient mineralization. Same comment for phosphate.

Author Response

Reviewer 2.

Summary: The authors carried out a study in which they investigated the impact of sound waves and microaeration (and their interaction) on anaerobic digestion of poultry litter slurry. They found that sound and microaeration had benefits to biogas production (similar to their past findings) but that there was minimal benefit of interaction. I found the study interesting but a little simplistic. Most of my comments are aimed at improving the clarity of the presentation.

Major comments:

  1. Any time an aeration rate or solids loading rate are reported, it needs to be reported as a specific (normalized by reactor volume) rate to have meaning beyond this study. This occurs throughout the manuscript e.g. L14-15, L17.

The aeration rate per digestate volume is now mentioned on Lines 113-114. The initial VS loading rate is now given on lines 140-141. Subsequent loading rates could be determined by simple math if desired.

  1. No description of analytical methods is provided; only a citation on L150. This forces the reader to go find the other paper to figure out what the authors did. Instead, provide a brief description of the methods and then cite if a reader wants to get more details.

We did this (Lines 152-159) and added a reference to American Public Health association methods. We renumbered subsequent references.

  1. More specifics are needed on statistical analysis – what tests were used? It should also be noted that averages were calculated across repeat measures from the same reactor – i.e. no biological replication. Thus measurements are not truly independent which is an assumption of most statistical tests. Researchers do this all the time due to practical limitations but probably need to point it out.

On lines 158-159, we pointed out that the means and comparisons were computed from repeated measurements over time.          

We gave more details of the statistical tests, (Lines 157-159) as well as added footnotes to tables 1&2 , and pointed out that the measurements were of repeated measurements over time. This is common in studies of digesters and treatment systems due to practical limitations as pointed out by the reviewer although it is not ideal by any means.

  1. The results and discussion jumps around quite a bit. Headers within this section would help indicate transitions to new topics.

We added some headers to make navigation easier.

  1. In Figure 1, I would like to see a second panel graph showing specific biogas production (or even better, CH4) per g VS as a function of loading rate. I suspect that microaeration increases overall biogas but at the expense of specific methane production. This is somewhat confirmed by the values on L197.

By looking at Table 1, if aeration alone is compared to the control, and aeration plus sound is compared to sound treatment alone it can be seen that aeration did not negatively affect methane production. Since the methane values are similar among all treatments, we don’t feel that adding a second panel to figure 1 would add needed information to the paper. This information can now be seen in the supplementary file, though.

  1. Table 1, the presentation of results is a little confusing. In the footer, it says values are the “mean of 61 summed weekly gas production.” I initially took this to mean that total cumulative biogas was summed over the entire period but realized this value is really just the average weekly biogas production across all different solids loading rates. This decision unfortunately hinders the statistical power of the analysis because different loading rates are expected to lead to different total production yet this variation is now part of the error term. Why not instead normalize all production (biogas, CO2, CH4) by the VS loading rate before taking the average? This would correct for the expected variation in biogas production in response to changing VS loading and reduce variability. I understand this rate will still vary with loading rate but I think this is more appropriate than the current presentation of data. In addition, doing so makes the result comparable to the literature since reporting of L CH4/g VS is a standard metric for AD. I also would expect that COD, TSS, soluble C, and soluble N are a function of loading rate and may benefit from normalization before taking an average across loading rates.

We changed Figure 1 to present the yield of methane versus the volatile solids loading rate. This along with table 1 gives the overall experimental yield plus the yield versus the loading rate so that the reader can see both. For convivence, we added a secondary x-axis showing the litter feeding rate.  

  1. Reporting methane and CO2 as ug/L makes it hard to compare to the literature. Normally a molar or volumetric (at STP) ratio is used.

We changed table 1 to report methane and carbon dioxide in molar units and figure 1 as millimoles of methane per week.

Minor comments:

L142: 400 g PL/week for 6 weeks?

L187: 0.35 ml/g VS does not look right. Should be 0.35 L/g VS. Also, this value will vary widely depending on substrate and digester conditions – the citation references general AD. I would encourage the authors to instead find other studies of poultry litter AD and report that number instead. It would be a much more fair comparison to the author’s work.

The reviewer is correct about the yield of methane. We apologize for the error.

Table 2: I’m surprised these ammonium levels are so low given the fairly high PL loadings. TN in Table 1 seems reasonable and is >10x higher than ammonium which makes me think this digester is not very effective at nutrient mineralization. Same comment for phosphate.

It is really hard to glean much from these numbers. The calcium and particularly magnesium concentrations are also a little low which makes me suspect struvite precipitation in the tanks. There may have been considerable NH4MgPO4 precipitation which may have kept the ammonium and phosphate concentrations in the wastewater low. The phosphate concentrations are higher in the beginning of the experiment which may support this line of reasoning. As the experiment progressed, the pH of the digesters increased from just above 6.5 to well above 7.0. This would favor the precipitation of struvite, removing some ammonium and especially PO4 from the digestate. But since we did nor perform analysis of the sludge, this is largely supposition.